# Patient knowledge in anaesthesia: Psychometric development of the RAKQ–The Rotterdam anaesthesia Knowledge questionnaire

**Sander F. van den Heuvel**[1]*, **Hester van Eeren**[2], **Sanne E. Hoeks**[1], **Anna Panasewicz**[3], **Philip Jonker**[1], **Sohal Y. Ismail**[2], **Jan J. van Busschbach**[2], **Robert Jan Stolker**[1], **Jan-Wiebe H. Korstanje**[1]

1 Department of Anaesthesiology, Erasmus MC, University Medical Centre Rotterdam, Rotterdam, The Netherlands, 2 Department of Psychiatry, Erasmus MC University Medical Centre Rotterdam, Rotterdam, The Netherlands, 3 Department of Anaesthesiology, Albert Schweitzer Hospital, Dordrecht, The Netherlands

* s.vandenheuvel@erasmusmc.nl

**Data Availability Statement:** All data files are available from the Dataverse database (accession number doi:10.34894/3UCXY4). A Data Transfer

## Abstract

The transition from in-person to digital preoperative patient education requires effective methods for evaluating patients' understanding of the perioperative process, risks, and instructions to ensure informed consent. A knowledge questionnaire covering different anaesthesia techniques and instructions could fulfil this need. We constructed a set of items covering common anaesthesia techniques requiring informed consent and developed the Rotterdam Anaesthesia Knowledge Questionnaire (RAKQ) using a structured approach and Item Response Theory. A team of anaesthetists and educational experts developed the initial set of 60 multiple-choice items, ensuring content and face validity. Next, based on exploratory factor analysis, we identified seven domains: General Anaesthesia–I (regarding what to expect), General Anaesthesia–II (regarding the risks), Spinal Anaesthesia, Epidural Anaesthesia, Regional Anaesthesia, Procedural sedation and analgesia, and Generic Items. This itemset was filled out by 577 patients in the Erasmus MC, Rotterdam, and Albert Schweitzer Hospital, Dordrecht, the Netherlands. Based on factor loadings (≥0.25) and considering clinical relevance this initial item set was reduced to 50 items, distributed over the seven domains. Each domain was processed to produce a separate questionnaire. Through an iterative process of item selection to ensure that the questionnaires met the criteria for Item Response Theory modelling, 40 items remained in the definitive set of seven questionnaires. Finally, we developed an Item Response Theory model for each questionnaire and evaluated its reliability. 1-PL and 2-PL models were chosen based on best model fit. No item misfit (S-$\chi^2$, p<0.001 = misfit) was detected in the final models. The newly developed RAKQ allows practitioners to assess their patients' knowledge before consultation to better address knowledge gaps during consultation. Moreover, they can decide whether the level of knowledge is sufficient to obtain digital informed consent without face-to-face education. Researchers can use the RAKQ to compare new methods of patient education with traditional methods.

Agreement (DTA) in line with GDPR regulations and/or a Research Collaboration Agreement (RCA) should be signed before data is shared.

**Funding:** This study was supported by a grant provided by the Dutch Ministry of Economic Affairs to JK (TKI grant No. EMCLSH200009, URL: https://www.health-holland.com/en/project/2021/2020/making-preoperative-anesthesiological-screening-future-proof). The sole responsibility for the content of this publication lies with the authors. The funders had no role in study design, data collection and analysis, decision to publish, or preparation of the manuscript. There was no additional external funding received for this study.

**Competing interests:** I have read the journal's policy and the authors of this manuscript have the following competing interests: JK was an unpaid medical adviser for NovaCair B.V., a developer of digital preoperative screening software. This does not alter our adherence to PLOS ONE policies on sharing data and materials.

# Introduction

Informed consent is legally required for any medical procedure or therapy. Preoperative patient education on the procedure, risks, and necessary preparations are essential for obtaining informed consent for any type of anaesthesia (e.g. general, spinal, epidural, or regional anaesthesia). Therefore, it is an integral part of preoperative anaesthetic consultations. Today's technological advancements make digital preoperative education and screening feasible, with less direct involvement of the anaesthetist, less need for hospital real estate, and a lower burden on patients in terms of time and costs spent on travelling [1]. To facilitate the transition from face-to-face education and screening to digital education and screening, there is a need to compare novel teaching methods, for example educating patients using video animation, to a conventional face-to-face setting in terms of patients' knowledge level on anaesthesia. A psychometrically validated knowledge questionnaire covering all aspects of anaesthesia could be a tool to evaluate patients' knowledge before obtaining informed consent and a first step in digitally screening patients.

Although several questionnaires have been developed to measure knowledge on anaesthetic topics [2–5], only the questionnaire developed by Miller et al. was validated [3]. However, it was intended for the parents of paediatric patients. Moreover, these earlier questionnaires covered only general anaesthesia and preoperative instructions. None of the questionnaires tested knowledge on other anaesthesia techniques. Furthermore, none of the questionnaires were validated for use in different populations, and none were developed for use in daily practice to test individual patients' knowledge.

To assess patients' knowledge on anaesthesia effectively and validly in a digital setting, new psychometrically validated questionnaires are needed for a wider variety of anaesthesia types. Therefore, we designed this study to construct a set of items covering the most common anaesthesia techniques that require informed consent (i.e. general anaesthesia, spinal anaesthesia, epidural anaesthesia, regional anaesthesia, procedural sedation and analgesia (PSA), and generic items). The primary objectives were to (i) develop a set of items covering knowledge on different anaesthesia techniques, (ii) develop scales along which this knowledge can be graded, and (iii) construct models using Item Response Theory (IRT) and determine the psychometric properties of the items.

# Materials and methods

Based on the methodology described by Boateng et al.[6] and Reeve et al.[7], we divided the process of developing knowledge scales into two distinct phases: I) scale development, and II) scale evaluation, with each phase having multiple steps, as shown in Fig 1. This study aimed to create a comprehensive set of questionnaires covering six distinct knowledge domains regarding anaesthesia: general anaesthesia, spinal anaesthesia, epidural anaesthesia, regional anaesthesia, PSA, and generic items (e.g. preoperative instructions). The development and adaptation of the questionnaires are explained in the following steps, adhering to psychometric and clinical guidelines. See S1 Table in the Online Supporting Information for a glossary of terms pertaining to the development of questionnaires.

## Phase I–scale development

**1. Item generation.** *A. Identification of domains and item generation.* First, we conducted a literature search to collect sample questions from published questionnaires on patient knowledge regarding anaesthesia to guide item generation. Two anaesthetists (SH and JK) then developed an item bank of 51 items covering the six aforementioned knowledge domains. The

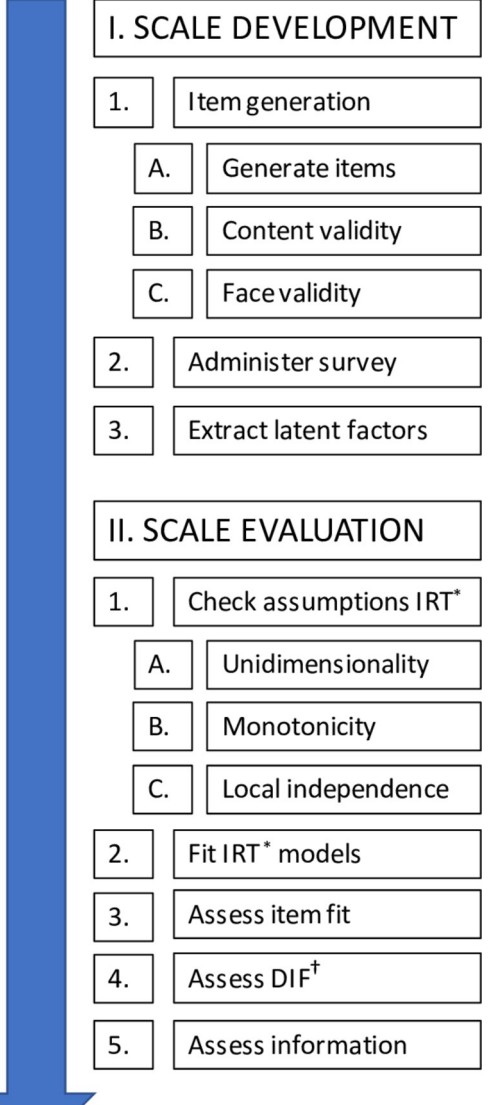

**Fig 1. Steps in developing RAKQ (adapted from Boateng et al.[7] and Reeve et al.[1]).** *Item Response Theory; †Differential item Functioning.

items were formulated as multiple-choice items with three to five answer categories, including an 'I do not know' option to discourage guessing. All the items had only one correct answer.

*B. Content validity*. To ensure content validity, all items were presented to 14 anaesthetists and three registrars in both a general hospital and an academic hospital. They were asked to accept or reject an item or suggest adaptation of an item. The resulting item bank was reviewed by an educational expert and psychologist, and the formulation of the items was adapted to ensure that language level B1 [8] was not exceeded and to prevent ambiguous answer categories.

*C. Face validity*. To ensure face validity, this set of items was presented to three patients visiting the outpatient preoperative screening clinic. After completing the questionnaires, the patients were interviewed about the appropriateness of the questionnaires to measure

knowledge on anaesthesia, and comprehension of all items was assessed. Their comments were used to adapt the items.

**2. Sampling and survey administration.** As a rule of thumb, the sample size for IRT analysis requires ten participants per item [9]. Therefore, the appropriate number of participants was calculated based on the resulting number of items generated during phase I-1. (Item generation). These participants were recruited at two hospital sites. Between September 2020 and November 2020, participants were included on the outpatient preoperative clinic of the Erasmus Medical Centre (Erasmus MC) in Rotterdam, the Netherlands, a university hospital. From April 2021 to May 2021, participants were included on the outpatient preoperative clinic of the Albert Schweitzer Hospital (ASZ) in Dordrecht, the Netherlands, a general teaching hospital. Patient inclusion criteria were a minimal age of 18 years, ability to read and understand Dutch, and planned elective surgery. Before administering the questionnaire in the ASZ, slight textual adjustments were made by a team of anaesthetists and a psychologist, based on remarks from participants in Erasmus MC. Care was taken not to change the content of the items or the answer categories (S2 Document in S1 File). Special attention on differential item functioning was paid to the altered items during the scale evaluation phase.

The questionnaires were sent by e-mail after the preoperative consultation had taken place. All participants were asked to complete all questionnaires, regardless of the anaesthesia technique they were educated on. Furthermore, the anaesthetists were not informed about the participation of their patients in this study, and the education provided was limited to anaesthesia techniques relevant to the planned surgery and in accordance to standard practice. The questionnaires were presented on a secure online survey platform.

Age, sex and the planned anaesthesia technique were extracted from the hospital information system during the period of inclusion. This data was pseudonymised and the authors did not have access to information that could identify individual participants.

**3. Extraction of factors.** To explore latent factors within each predefined anaesthesia domain, we performed exploratory factor analysis using Multidimensional IRT (MIRT). To determine the number of latent factors that best described the data, we first fitted multiple MIRT models with an increasing number of factors using the Metropolis-Hastings Robbins-Monro algorithm [10]. We compared nested models using the Akaike information criterion (AIC), Bayesian Information Criteria (BIC), and chi-square difference tests. Although we took into account the BIC, in this exploratory phase we preferred the AIC over the BIC to decide which model to choose, because the AIC applies a smaller penalty than the BIC and thus reduces the risk of losing too much information.

Next, we explored the factor loadings in the optimal MIRT model option for each domain. To allow the factors to correlate, we used oblique rotation (Oblimin) [11]. The exploratory nature of the analysis and the large sample size made us consider items with factor loadings as low as 0.25 [12]. Guided by the factor loadings, we assembled item sets based on the following: factors needed to be conceptually interpretable, the first factor was prioritised above the next (since every next factor is extracted based on the residual of the previous factor), and a minimum of three items per factor was needed (to facilitate further analysis) [11]. If the item contributed conceptually to one of the factors, the cross-loading of an item on multiple factors was not used as a reason for removal. When an item did not load on the factor that was selected for the final item set, providing it was conceptually fitting that factor, it was deemed relevant from a clinical perspective and there were no doubts about the quality of the item, the item was kept in the item set during this phase of the scale development. Each selected factor represented a separate scale, corresponding to a separate questionnaire, and was evaluated in the next phase.

## Phase II–scale evaluation

**1. Checking of assumptions IRT.** Answer categories were dichotomised into correct vs. incorrect answers, where the category 'I don't know' was categorised as being incorrect. To fit an IRT model the assumptions of unidimensionality, local independence, and monotonicity must be met [7]. We assessed these assumptions for each scale, derived in phase I based on item generation and scale development. The assumptions were tested in an iterative process in which an assumption was tested again if the next assumption was not met for that scale.

*A. Unidimensionality.* Confirmatory factor analysis with weighted least square mean- and variance-adjusted estimator was performed to assess the unidimensionality of each scale. Unidimensionality was accepted when the following criteria were met: a standardised root mean square residual (SRMR) value < 0.08, root mean square error of approximation (RMSEA) value < 0.06 and Scaled Comparative Fit Index (CFI) and Tucker-Lewis Index (TLI) values > 0.95 [13–18]. When unidimensionality was not satisfactory, items were chosen to be left out in subsequent analysis, based on factor loading and clinical relevance. The final set of items in each scale was subjected to an additional check for unidimensionality with Modified Parallel Analysis [19].

*B. Monotonicity.* We assessed monotonicity and scalability and invariant item ordering (IIO) by fitting nonparametric IRT functions using Mokken scaling [20]. The assumption of monotonicity was considered met when no significant violations of monotonicity were detected based on the z-test statistics. Scalability was assessed using the value of the scalability coefficient H of the entire scale and individual items ($H_i$). The H value of the entire scale was considered strong if $H > 0.50$, moderate if $0.40 \leq H \leq 0.50$ and weak if $0.30 \leq H \leq 0.40$ [21]. IIO was determined by assessing the number of significant violations of IIO per item based on the z-test statistic. When significant violations of monotonicity or IIO were detected or unscalable items were encountered, backward selection was applied by stepwise removal of the items with the highest violations or the lowest H coefficient until no violations were detected. This process ended when the H-coefficient of the entire scale was satisfactory [22].

*C. Local independence.* Local independence was met if the residual correlations between all the item pairs in the CFA model were < 0.20 [7]. When local dependencies were detected, we reviewed those items to determine the nature of the dependency and removed them when deemed necessary.

**2. IRT model fitting.** Once the assumptions were met, the item response functions, that is 1-PL, 2-PL, and 3-PL models, were fitted per scale, using the Expectation Maximization estimation algorithm. Nested models were compared using a likelihood ratio test, AIC, and BIC, and the best-fitting model was chosen [23]. The goodness of fit of the IRT models was assessed at item level using the S-$\chi^2$ statistic for item fit, for which a p-value of <0.001 for an item was considered a misfit [7, 24, 25].

**3. Differential item functioning.** We evaluated Differential Item Functioning (DIF) using ordinal regression modelling with McFadden's pseudo $R^2 \geq 0.02$ being indicative of DIF ($R^2_{12}$ suggests uniform DIF, $R^2_{23}$ suggests non-uniform DIF) [26]. We evaluated age (two groups divided by the median age of the study sample), sex, the hospital the patient visited for preoperative screening (EMC vs. ASZ), level of education (tertiary vs. primary, secondary, and other), and whether the anaesthesia technique discussed with the participant during the preoperative consultation matched the knowledge domain of the respective scale (yes vs. no). To assess the magnitude and statistical relevance of DIF when an item was flagged for DIF, we compared the difference between the initial theta's (i.e. the estimated level of ability) and the theta's without items with DIF (i.e. purified theta's) on a scale level. This difference was then

evaluated by plotting the differences due to DIF against the median standard error of measurement (SEM) [27]. Differences larger than the mean SEM were considered noticeable.

**4. Information.** In IRT, each response pattern results in a different theta and a different associated standard error. Therefore, the precision or reliability of an IRT model differs across the range of theta and is conceptualised as information. To investigate the information of the models, Test Information Curves were plotted together with the SE(theta), which was estimated using the Expected A Posteriori estimator. When compared with classical testing, an SE (theta) of 0.32 or lower, corresponds to a reliability of 0.90 or higher, and can thus be considered a reliable measurement [28].

### Statistics and data analysis

Data were collected using LimeSurvey [29] and Gemstracker [30]. Data analysis was performed using R (version 4.2.0) [31]. Unidimensionality and local independence were assessed with the R package 'lavaan' [32] and 'ltm'[33]. Monotonicity, scalability and invariant item ordering (IIO) were assessed with the R-package 'mokken' [21, 34]. Differential Item Functioning (DIF) was evaluated with the R-package 'lordif' [35]. Exploratory Factor Analysis, IRT modelling and assessment of the test information were performed with the R-package 'mirt' [23, 36].

Demographic and clinical characteristics were compared between the two hospitals and expressed as mean (SD) or number (percentage), where appropriate. Continuous variables were compared using Welch's t-test, Mann-Whitney U test or Kruskal-Wallis test where appropriate, categorical variables were compared using the Chi-Square test.

### Ethical considerations

Ethical approval for this study was granted by the Medical Ethics Committee of Erasmus MC Rotterdam (MEC-2020-0468). Because there was no infringement on the physical and/or psychological integrity of the subject, the Medical Ethics Committee deemed the trial not to be subject to the Dutch Law on Medical Research [37]. Written informed consent was obtained from all subjects. This study was conducted in compliance with the principles of the Declaration of Helsinki [38].

## Results

In this study, we developed sets of items covering knowledge on different anaesthesia techniques and constructed scales in which this knowledge can be graded. We then constructed models using IRT and determined the psychometric properties of the items.

### Phase I–scale development

**1. Item generation.** Initially, six knowledge domains were defined: Generic items, General anaesthesia, Spinal anaesthesia, Epidural anaesthesia, Regional anaesthesia and PSA. Then, 51 items covering these knowledge domains were formulated, which were subsequently supplemented by 9 items following the review of a larger group of experts, in total 60 items. The full list of 60 items, as a non-validated English translation of the Dutch original, that resulted after checking for content and face validity is shown in Table 1. Additionally, this table indicates the phase at which each item was removed from the item set and the reason for its removal.

**2. Sampling and survey administration.** Given the fact 60 items were generated during phase I-1 (Item generation), a sample size of 600 participants would be needed, which was

**Table 1. List of items resulting from Phase I-1 –Item generation.**

| Item ID | Item stem | Phase of item removal | Reason for removal |
|---|---|---|---|
| *Generic items* | | | |
| GEN6 | Why do you need to be fasted for every type of anaesthesia? | I-3. Factor extraction | No factor loading, quality of item unsatisfactory |
| GEN7 | How long before the operation are you allowed to drink clear fluids? | I-3. Factor extraction | Too few items loading on factor, no contribution to construct |
| **GEN8** | **What must you do with your usual daily medications?** | | |
| GEN9 | General anaesthesia is safer if you stop smoking a few weeks before the operation. | I-3. Factor extraction | No factor loading, quality of item unsatisfactory. |
| GEN10* | You may drink milk 6 hours before any form of anaesthesia (general anaesthesia, regional anaesthesia, spinal anaesthesia). | III-1. Scalability | Low scalability ($H_i$) |
| **GEN11*** | **Which of the fluids stated below may you drink up to 2 hours before the operation?** | | |
| GEN12 | May you keep your dentures/false teeth in place during any form of anaesthesia (general anaesthesia, regional anaesthesia, spinal anaesthesia)? | II-1. Scalability | Low scalability ($H_i$) |
| GEN1 | An anaesthetist is a. . . | I-3. Factor extraction | Too few items loading on factor, conceptually not fitting |
| **GEN2*** | **Patients are generally seen in advance of the surgery by the same anaesthetist who administers the anaesthesia on the day of the surgery.** | | |
| **GEN3** | **What can anaesthetists do to reduce anxiety?** | | |
| GEN4 | The patient must give permission in advance for every type of anaesthesia (general anaesthesia, regional anaesthesia, spinal anaesthesia). | II-1. Scalability | Low scalability ($H_i$) |
| GEN5 | Before the operation, the anaesthetist determines the possible risks for an individual patient during anaesthesia (general anaesthesia, regional anaesthesia, spinal anaesthesia). | I-3. Factor extraction | Too few items loading on factor, conceptually not fitting |
| *General anaesthesia* | | | |
| GA1 | When you receive general anaesthesia, you have to be ventilated (attached to a breathing machine). | II-1. Invariant Item Ordering | Violation of IIO |
| **GA2*** | **What do you notice from the breathing tube placed in your mouth during the operation?** | | |
| **GA3*** | **How does an anaesthetist administer medication that puts a person under anaesthesia?** | | |
| **GA4** | **Where do most patients wake up after general anaesthesia?** | | |
| **GA5** | **Who administers general anaesthesia?** | | |
| **GA6** | **Under general anaesthesia, it is possible to develop a damaged nerve from being in a particular position for too long.** | | |
| **GA7** | **To insert the breathing tube, the anaesthetist makes a small incision in the wind pipe and reseals the opening at the end of the operation.** | | |
| GA8* | Patients usually wake up rapidly after the operation. | I-3. Factor extraction | Factor conceptually not interpretable, quality of item unsatisfactory |
| **GA9** | **When is the breathing tube placed?** | | |
| **GA10*** | **Sometimes, your teeth can be damaged during the placement of the breathing tube.** | | |
| GA11 | How does the anaesthetist know that the patient feels no pain while under general anaesthesia? | II-1. Invariant Item Ordering | Violation of IIO |
| **GA12** | **Does every patient experience nausea after general anaesthesia?** | | |
| GA13 | Many patients wake up unintentionally during an operation. | II-1. Unidimensionality | Borderline model fit, quality of item unsatisfactory |
| **GA14** | **Patients must tell the anaesthetists if they have loose teeth.** | | |
| **GA15** | **Can general anaesthesia cause dementia?** | | |
| **GA16** | **After an operation under general anaesthesia, a patients ability to concentrate can be reduced for a short period of time.** | | |
| *Spinal anaesthesia* | | | |
| **SA1** | **What part of the body is numbed during spinal anaesthesia?** | | |
| **SA2** | **What are the possible side effects of spinal anaesthesia?** | | |

*(Continued)*

**Table 1.** (Continued)

| Item ID | Item stem | Phase of item removal | Reason for removal |
|---|---|---|---|
| SA3 | The patient can always observe the entire operation during spinal anaesthesia. | II-1. Invariant Item Ordering | Violation of IIO |
| **SA4** | **A spinal injection is more painful than a regular injection in the arm.** | | |
| **SA5** | **How long before spinal anaesthesia wears off?** | | |
| **SA6** | **Are mild tingling sensations normal during spinal anaesthesia placement?** | | |
| **SA7** | **Important vital signs (e.g. blood pressure and heart rate) are measured during spinal anaesthesia, similar to general anaesthesia.** | | |
| SA8 | Is it possible to urinate while under spinal anaesthesia? | I-3. Factor extraction | Too few items loading on factor, quality of item unsatisfactory |
| **SA9** | **During an operation under spinal anaesthesia, it is possible to also receive sedation.** | | |
| SA10 | Permanent paralysis after spinal anaesthesia (numb from the waist down) is very rare. | II-1. Invariant Item Ordering | Violation of IIO |
| **SA11** | **Patients are discharged home faster after spinal anaesthesia than after general anaesthesia.** | | |
| **SA12** | **After the operation, as soon as feeling and strength have returned to the legs after spinal anaesthesia, patients are allowed to drive home themselves.** | | |
| **SA13** | **What action does the anaesthetist take if spinal anaesthesia fails after one or more attempts?** | | |
| *Regional anaesthesia* | | | |
| RA1 | Regional anaesthesia works immediately after the injection of the local anaesthetic medication. | I-3. Factor extraction | Too few items loading on factor, quality of item unsatisfactory |
| **RA2** | **Which tool can anaesthetists use to administer regional anaesthesia?** | | |
| **RA3** | **Regional anaesthesia can work for longer than 12 hours.** | | |
| RA4 | A patient may only be discharged from the hospital once the effects of regional anaesthesia have worn off. | II-1. Scalability | Low scalability ($H_i$) |
| RA5 | After regional anaesthesia, patients can sometimes experience pins and needles in the limb during the first few weeks to months after the operation. | II-1. Scalability | Low scalability ($H_i$) |
| **RA6** | **During regional anaesthesia, the affected limb is not only numb, but the patient is also unable to move it.** | | |
| **RA7** | **The chance of nausea after regional anaesthesia is lower than that after general anaesthesia?** | | |
| *Epidural anaesthesia* | | | |
| **EA1** | **The tube for pain medication is removed immediately after the operation.** | | |
| EA2 | After the operation, the patient is only allowed to sleep on his/her side because of the tube in their back. | I-3. Factor extraction | Too few items loading on factor, quality of item unsatisfactory |
| **EA3** | **Permanent nerve damage after epidural anaesthesia is very rare.** | | |
| **EA4** | **What is the advantage of epidural anaesthesia over the use of pain medication in drips?** | | |
| **EA5** | **If a patient with epidural anaesthesia cannot move their legs, what must he/she do?** | | |
| **EA6** | **What are the options if placement of the epidural was not successful?** | | |
| *Procedural sedation and analgesia* | | | |
| **PSA1** | **During sedation, as with general anaesthesia, the patient stops breathing spontaneously.** | | |
| **PSA2** | **After sedation, patients can sometimes remember some occurrences from during the sedated period.** | | |
| **PSA3** | **The patient must be fasted for a procedure performed under sedation.** | | |
| PSA4 | Sedation can be carried out by an anaesthesia nurse with special training. | I-3. Factor extraction | Too few items loading on factor, quality of item unsatisfactory |
| **PSA5** | **What is true for both sedation and general anaesthesia?** | | |

(*Continued*)

**Table 1.** (Continued)

| Item ID | Item stem | Phase of item removal | Reason for removal |
|---------|-----------|----------------------|-------------------|
| **PSA6** | **Are there procedures for which sedation is not enough and general anaesthesia is required?** | | |

In bold the items included in the scale after phase I and II. Reasons for removal of the item from the set are given. * Items with changes in wording between Erasmus MC and Albert Schweitzer Hospital. See online Supporting Information S2 Document in S1 File.

approximated by including 577 patients visiting the preoperative outpatient clinic. In the Erasmus MC, 610 patients consented to be approached regarding this study. The response rate was 55% (n = 336) and of these respondents, 95% provided informed consent for inclusion and completed the questionnaire (n = 319). In ASZ, 950 patients consented to be approached regarding this study. The response rate was 39% (n = 370) and of these respondents, 70% provided informed consent for inclusion and completed the questionnaire (n = 258).

Table 2 shows the demographic and clinical characteristics of the in total 577 participants who completed the questionnaire. All participants answered all items in the questionnaire. Across all participants, the mean age was 60 (± 15) and 274 (48%) were female. The percentage of female participants was 41% versus 56% in Erasmus MC and ASZ, respectively (p = 0.002). In ASZ, spinal and regional anaesthesia was planned more often compared to Erasmus MC (20% versus 2% and 10% versus 3%, respectively, p<0.001). Other types of planned anaesthesia techniques, as well as age and highest level of education did not differ significantly between participants.

**3. Extraction of factors.** Table 3 shows the factor loadings >0.25 for all items within each predefined knowledge domain, based on the optimal number of factors as determined by model comparison (S2 Table). Items retained in the final item set(s) are printed in bold. The domains Regional anaesthesia, Spinal anaesthesia, Epidural anaesthesia and PSA all resulted in two factors, but were reduced to a single set of items per domain, including several items that did not load on the first factor, because of their clinical and conceptual relevance. The Generic

**Table 2. Demographics and clinical characteristics of the two sample sites.**

| | EMC* | ASZ† | Total | P-value |
|---|---|---|---|---|
| | *n = 319* | *n = 258* | *n = 577* | |
| Age; y | 59.3 (15.4) | 60.1 (14.7) | 59.7 (15.1) | 0.910 |
| Sex; female | 130 (41) | 144 (56) | 274 (48) | 0.002 |
| Highest level of education | | | | 0.252 |
| Primary | 18 (6) | 13 (5) | 31 (5) | |
| Secondary | 185 (58) | 160 (64) | 345 (61) | |
| Higher | 102 (32) | 78 (31) | 180 (31) | |
| 'I don't know' | 3 (1) | 0 (0) | 3 (1) | |
| Other | 11 (3) | 0 (0) | 11 (2) | |
| Anesthesia technique | | | | |
| General anaesthesia | 244 (77) | 190 (74) | 434 (75) | 0.694 |
| Spinal anaesthesia | 6 (2) | 52 (20) | 58 (10) | <0.001 |
| Epidural anaesthesia | 22 (7) | 14 (5) | 36 (6) | 0.701 |
| Regional anaesthesia | 9 (3) | 25 (10) | 34 (6) | 0.002 |
| Procedural sedation and analgesia | 62 (20) | 33 (13) | 95 (17) | 0.098 |

Values are mean (SD) or number of patients (valid percentage). * Erasmus MC Rotterdam; †Albert Schweitzer Hospital Dordrecht

**Table 3. Factor loadings for exploratory factor analysis.**

| Generic items | Factor 1 | Factor 2 | Factor 3 | | Regional anaesthesia | Factor 1 | Factor 2 |
|---|---|---|---|---|---|---|---|
| **GEN11** | **0.707** | | 0.439 | | **RA6** | **0.822** | |
| **GEN10** | **0.581** | | -0.311 | | **RA7** | **0.457** | |
| **GEN3** | **0.543** | 0.266 | | | **RA2** | **0.444** | 0.253 |
| **GEN8** | **0.483** | | | | **RA3** | **0.384** | 0.417 |
| **GEN4** | **0.345** | | | | **RA5** | **0.383** | |
| **GEN12** | **0.252** | | | | RA4 | * | 0.827 |
| **GEN2** | * | | | | RA1 | | 0.267 |
| GEN5 | | 0.760 | | | | | |
| GEN1 | | 0.665 | | | | | |
| GEN7 | | | 0.905 | | | | |
| GEN6 | | | | | | | |
| GEN9 | | | | | | | |

| General anaesthesia | Factor 1 | Factor 2 | Factor 3 | Factor 4 | Spinal anaesthesia | Factor 1 | Factor 2 |
|---|---|---|---|---|---|---|---|
| **GA9** | **0.890** | | | | **SA6** | **0.938** | |
| **GA3** | **0.647** | | | | **SA7** | **0.757** | |
| **GA2** | **0.597** | | 0.263 | | **SA12** | **0.752** | |
| **GA7** | **0.530** | 0.302 | | | **SA9** | **0.735** | |
| GA10 | 0.528 | | | **0.392** | **SA13** | **0.661** | |
| **GA5** | **0.430** | | | 0.402 | **SA10** | **0.626** | |
| **GA1** | **0.411** | | | 0.288 | **SA4** | **0.620** | |
| **GA11** | **0.387** | | | | **SA5** | **0.488** | 0.327 |
| GA12 | 0.292 | | 0.547 | * | **SA3** | **0.443** | |
| GA14 | 0.263 | | 0.283 | **0.442** | **SA2** | **0.428** | 0.336 |
| **GA4** | **0.262** | | | | **SA11** | **0.355** | 0.312 |
| GA13 | | 0.867 | | * | SA8 | 0.286 | 0.577 |
| GA15 | | 0.637 | 0.292 | * | **SA1** | * | 0.944 |
| GA8 | | | 0.492 | | | | |
| **GA6** | | | | **0.814** | | | |
| **GA16** | | | | **0.268** | | | |

| Epidural | Factor 1 | Factor 2 | | | Procedural sedation and analgesia | Factor 1 | Factor 2 |
|---|---|---|---|---|---|---|---|
| **EA6** | **0.795** | | | | **PSA2** | **0.765** | |
| **EA4** | **0.771** | | | | **PSA1** | **0.638** | |
| **EA5** | **0.468** | | | | **PSA6** | **0.616** | |
| **EA3** | **0.423** | 0.339 | | | **PSA5** | **0.447** | |
| **EA1** | * | 0.747 | | | **PSA3** | * | 0.802 |
| EA2 | | 0.870 | | | PSA4 | | 0.522 |

Bold items were included for further analyses. Items with an asterisk (*) did not load on the factor, but were included based on conceptual fit to another factor.

items domain consisted of three factors and was reduced to a single set of items, because factors 2 and 3 consisted of too few items after allocating cross-loading items to factor 1. The General anaesthesia domain was reduced to two sets of items. Conceptually, factor 1 consisted of items relating to direct perioperative care ('General anaesthesia–I'), and factor 4 comprised of items concerning complications or side-effects ('General anaesthesia–II'). Factor 2 within General anaesthesia consisted of too few items when cross-loading was taken into account, and factor 3 was conceptually not interpretable. Items from factors 2 and 3 that fitted the construct of factor 4 were included in the 'General anaesthesia–II' item set for further analysis. In

**Table 4. Final assessment of Unidimensionality before IRT modelling can be undertaken.**

| Scale | Unidimensionality | | | |
|---|---|---|---|---|
| | SRMR* | RMSEA† | Scaled CFI‡ | Scaled TLI¶ |
| Generic items | 0.015 | 0.000 | 1 | 1.034 |
| General anaesthesia—I | 0.072 | 0.029 | 0.993 | 0.988 |
| General anaesthesia—II | 0.066 | 0.038 | 0.973 | 0.956 |
| Spinal anaesthesia | 0.061 | 0.045 | 0.982 | 0.977 |
| Regional anaesthesia | 0.018 | 0.000 | 1.000 | 1.027 |
| Epidural anaesthesia | 0.019 | 0.000 | 1 | 1.014 |
| Procedural sedation and analgesia | 0.050 | 0.044 | 0.977 | 0.954 |

* Standardized Root Mean Squared Residual; † Root Mean Square Error of Approximation; ‡ Comparative Fit Index; ¶ Tucker-Lewis Index

Table 1, the last column indicates the rationale for the exclusion of particular items from the item set. In total 7 factors were deemed suitable for further evaluation in the next phase as separate scales.

## Phase II–scale evaluation

**1. Checking of assumptions IRT.** The assessment of IRT assumptions and consecutive adjustments of the item sets is an iterative process which is elaborated on below. The final assessment of all IRT assumptions is presented in Tables 4–6. The scree plots following Modified Parallel Analysis are presented in S1 Fig in the Online Supporting Information, showing unidimensionality on all scales.

**Generic items** The initial set of seven items met the unidimensionality criteria, but was not scalable (H = 0.202; all items $H_i<0.3$). To improve scalability, the items GEN10, GEN4, and GEN12 were removed. The reduced set of four items met the unidimensionality criteria and was evaluated as weakly scalable (H = 0.307; two items $H_i<0.3$). No serious violations of monotonicity, violations of IIO or local dependencies were detected.

**General anaesthesia–I** The initial set of eight items met the unidimensionality criteria and was weakly scalable (H = 0.329; two items $H_i<0.3$). Analysis of IIO revealed that after removing two items (GA1 and GA11), no violation of IIO was detected. The reduced set of six items met the unidimensionality criteria and was moderately scalable (H = 0.441; no item $H_i <0.3$). No serious violations of monotonicity were detected. Two items showed local dependence

**Table 5. Final assessment of Monotonicity before IRT modelling can be undertaken.**

| Scale | Monotonicity | | | |
|---|---|---|---|---|
| | Scalability coefficients H | Scalability coefficients $H_i$ | Violations monotonicity | Violations IIO* |
| Generic items | 0.307 | 0.27–0.35 | none | none |
| General anaesthesia—I | 0.441 | 0.32–0.47 | none | none |
| General anaesthesia—II | 0.329 | 0.30–0.43 | none | none |
| Spinal anaesthesia | 0.439 | 0.37–0.61 | 2, none critical | none |
| Regional anaesthesia | 0.349 | 0.28–0.45 | none | none |
| Epidural anaesthesia | 0.391 | 0.28–0.44 | none | none |
| Procedural sedation and analgesia | 0.316 | 0.22–0.36 | none | none |

Violations with a crit value >40 were deemed critical.

* Invariant Item Ordering

**Table 6. Final assessment of Local Independence before IRT modelling can be undertaken.**

| Scale | Local Independence |
|---|---|
| | Residual correlation |
| Generic items | none |
| General anaesthesia—I | GA4 & GA5 (0.267) |
| General anaesthesia—II | none |
| Spinal anaesthesia | none |
| Regional anaesthesia | none |
| Epidural anaesthesia | none |
| Procedural sedation and analgesia | none |

(GA4 and GA5), but apart from being very easy questions, no clinically relevant correlation could be found, and they were preserved in the item set.

**General anaesthesia–II** The initial set of seven items did not meet the unidimensionality criteria (SRMR = 0.086). After reviewing the three items that did not load on the latent factor in Phase II and removing the clinically least relevant item (GA13), the reduced set of six items met the unidimensionality criteria. The item-set was weakly scalable (H = 0.329; no item $H_i < 0.3$). No serious violations of monotonicity, violations of IIO or local dependencies were detected.

**Spinal anaesthesia** The initial set of 12 items met the unidimensionality criteria and was moderately scalable (H = 0.410; one item $H_i < 0.3$). Analysis of IIO revealed that after removing two items (SA3 and SA10), no violation of IIO was detected. The reduced set of ten items met the unidimensionality criteria and was moderately scalable (H = 0.439; no item $H_i < 0.3$). No serious violations of monotonicity or local dependencies were detected.

**Regional anaesthesia** The initial set of six items met all but one unidimensionality criterion (scaled TLI) but was not scalable (H = 0.252; four items $H_i < 0.3$). To improve scalability, items RA4 and RA5 were removed. The reduced set of four items met the unidimensionality criteria and was weakly scalable (H = 0.349; one item $H_i < 0.3$). No serious violations of monotonicity, violations of IIO or local dependencies were detected.

**Epidural anaesthesia** The initial set of five items met the unidimensionality criteria and was weakly scalable (H = 0.391; one item $H_i < 0.3$). No serious violations of monotonicity, violations of IIO or local dependencies were detected.

**Procedural sedation and analgesia** The initial set of five items met the unidimensionality criteria and was weakly scalable (H = 0.316; four items $H_i < 0.3$). No serious violations of monotonicity, violations of IIO or local dependencies were detected.

**2. IRT model fitting.** Table 7 shows the item parameters for the 1-PL and 2-PL models fitted to the data. S3 Table shows comparisons between the 1-PL, 2-PL, and 3-PL models for each scale. The 1-PL model was the best fitting model for the Generic items, General anaesthesia–II and Regional anaesthesia scales. The 2-PL model was the best fitting model in the General anaesthesia–I, Spinal anaesthesia, Epidural anaesthesia, and PSA scales. Table 7 also shows the item fit per scale for the best fitting model. No item misfit was detected. Fig 2. shows the item characteristic curves (ICC) per scale for the best fitting model.

**3. Differential item functioning.** Table 8 shows which items have DIF and the magnitude and type of DIF (only McFadden's pseudo $R^2 \geq 0.02$ is shown). Overall, no items were flagged for DIF regarding sex, seven items were flagged for DIF regarding age, one item was flagged for DIF regarding anaesthesia technique, four items were flagged for DIF regarding the sample hospital, and four items were flagged for DIF regarding the level of education. The impact of DIF on the total scores for each scale was well below the mean SEM, and its impact was

**Table 7. IRT-model item parameters and item fit (p-values).**

| Item ID | IRT-model item parameters | | Item fit | |
|---|---|---|---|---|
| | 1-PL | 2-PL | $p(S-\chi^2)$ | |
| | Difficulty | Difficulty | Discrimination | |
| *Generic items** | | | | |
| GEN2 | 0.224 | 0.228 | 0.871 | 0.035 |
| GEN3 | -1.378 | -0.936 | 1.693 | 0.143 |
| GEN8 | -2.201 | -1.839 | 1.175 | 0.406 |
| GEN11 | -1.984 | -1.496 | 1.386 | 0.210 |
| *General anaesthesia–I†* | | | | |
| GA2 | -1.482 | -0.743 | 2.075 | 0.533 |
| GA3 | -2.999 | -1.594 | 1.822 | 0.208 |
| GA4 | -4.354 | -3.232 | 1.072 | 0.658 |
| GA5 | -5.214 | -2.913 | 1.694 | 0.732 |
| GA7 | -1.828 | -0.925 | 2.025 | 0.759 |
| GA9 | -1.495 | -0.677 | 2.849 | 0.588 |
| *General anaesthesia–II** | | | | |
| GA6 | 2.517 | 2.304 | 1.045 | 0.079 |
| GA10 | -0.770 | -0.601 | 1.341 | 0.105 |
| GA14 | -2.487 | -1.819 | 1.478 | 0.665 |
| GA16 | -1.365 | -1.014 | 1.454 | 0.605 |
| GA12 | -1.592 | -1.242 | 1.331 | 0.142 |
| GA15 | 0.244 | 0.261 | 0.813 | 0.105 |
| *Spinal anaesthesia†* | | | | |
| SA1 | 0.353 | 0.212 | 1.444 | 0.467 |
| SA2 | 1.391 | 0.862 | 1.499 | 0.137 |
| SA4 | 1.439 | 0.812 | 1.838 | 0.431 |
| SA5 | 1.081 | 0.606 | 1.881 | 0.036 |
| SA6 | 0.775 | 0.408 | 2.408 | 0.173 |
| SA7 | -1.402 | -0.729 | 2.209 | 0.075 |
| SA9 | 0.029 | 0.023 | 2.007 | 0.119 |
| SA11 | 2.282 | 1.681 | 1.146 | 0.287 |
| SA12 | -1.085 | -0.700 | 1.438 | 0.125 |
| SA13 | -0.097 | -0.062 | 1.649 | 0.549 |
| *Regional anaesthesia** | | | | |
| RA2 | 1.833 | 1.467 | 1.236 | 0.595 |
| RA3 | 1.674 | 1.509 | 1.034 | 0.543 |
| RA6 | 0.476 | 0.322 | 1.683 | 0.149 |
| RA7 | -1.246 | -1.029 | 1.179 | 0.115 |
| *Epidural anaesthesia†* | | | | |
| EA1 | 2.506 | 1.623 | 1.537 | 0.209 |
| EA3 | -0.101 | -0.069 | 1.521 | 0.124 |
| EA4 | 0.493 | 0.287 | 1.908 | 0.331 |
| EA5 | 1.581 | 1.488 | 0.858 | 0.193 |
| EA6 | 0.709 | 0.383 | 2.380 | 0.407 |
| *Procedural sedation and analgesia†* | | | | |
| PSA1 | -1.040 | -0.683 | 1.835 | 0.560 |
| PSA2 | 0.702 | 0.536 | 1.367 | 0.157 |
| PSA3 | -1.398 | -1.657 | 0.733 | 0.748 |

*(Continued)*

**Table 7.** (Continued)

| Item ID | IRT-model item parameters | | | Item fit | |
|---|---|---|---|---|---|
| | 1-PL | 2-PL | | p(S-$\chi^2$) | |
| | Difficulty | Difficulty | Discrimination | | |
| PSA5 | 0.683 | 0.566 | 1.195 | 0.058 | |
| PSA6 | -1.850 | -1.470 | 1.277 | 0.546 | |

\* Item fit for 1-PL model; † Item fit for 2-PL model

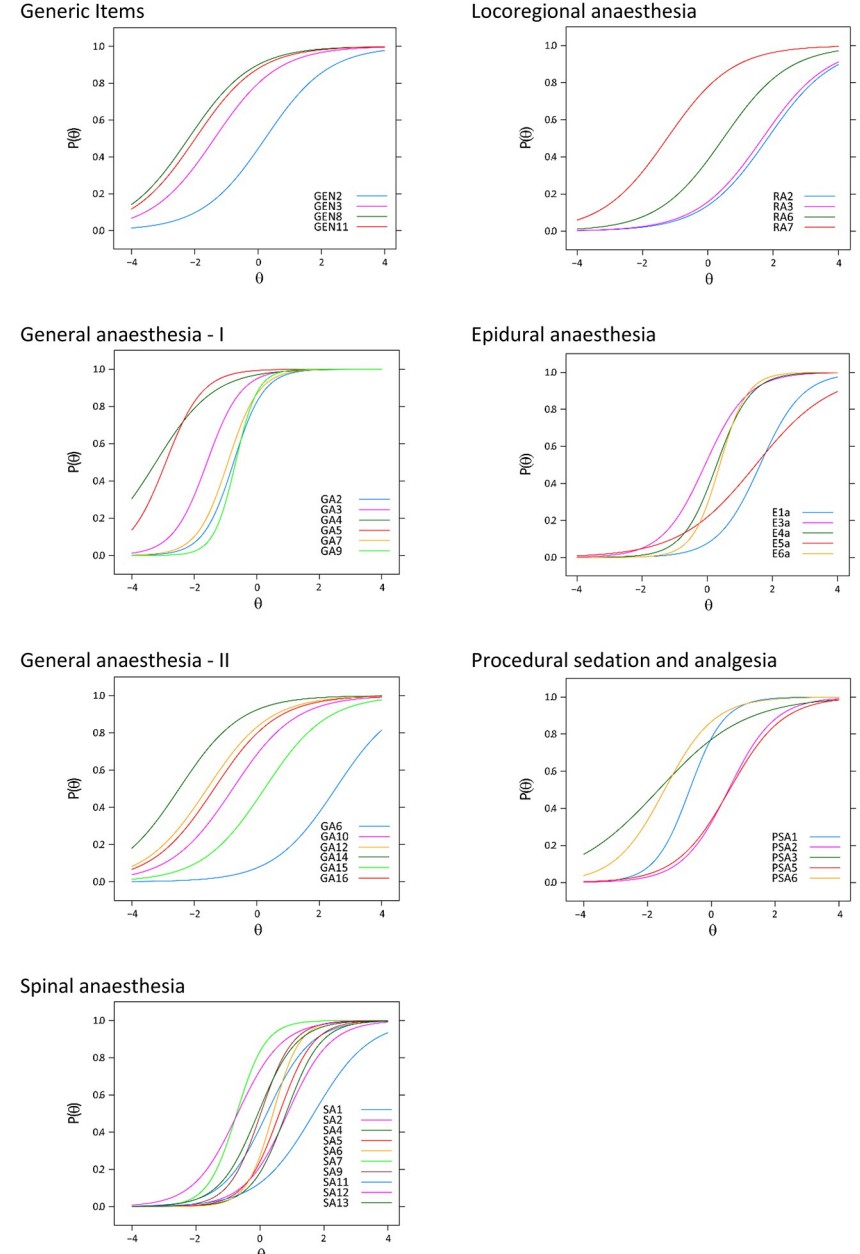

**Fig 2. Item response functions per scale.**

**Table 8. Differential item functioning (pseudo-$R^2$) per model.**

| Item ID | Differential Item Functioning | | | | | | | | | |
|---|---|---|---|---|---|---|---|---|---|---|
| | Sex | | Age | | Technique | | Hospital | | Education | |
| | $R^2_{12}$ | $R^2_{23}$ | $R^2_{12}$ | $R^2_{23}$ | $R^2_{12}$ | $R^2_{23}$ | $R^2_{12}$ | $R^2_{23}$ | $R^2_{12}$ | $R^2_{23}$ |
| *Generic items* | | | | | | | | | | |
| GEN3 | | | | 0.106 | | | | | | |
| GEN11 | | | | 0.141 | | | | | | |
| *General anaesthesia—I* | | | | | | | | | | |
| GA2 | | | 0.034 | | | | | | | |
| GA4 | | | 0.022 | 0.030 | | | | | 0.030 | 0.028 |
| GA5 | | | 0.076 | | | | | | | |
| *General anaesthesia—II* | | | | | | | | | | |
| GA6 | | | | | | | | | | 0.073 |
| GA10 | | | | | | | 0.124 | | | |
| GA14 | | | | | | | | | | 0.064 |
| GA16 | | | | | | | | 0.040 | | |
| GA12 | | | | | | | | 0.032 | | |
| *Spinal anaesthesia* | | | | | | | | | | |
| SA11 | | | 0.055 | | | | | | | |
| *Regional anaesthesia* | | | | | | | | | | |
| RA2 | | | 0.037 | | | | | | | |
| *Epidural anaesthesia* | | | | | | | | | | |
| EA1 | | | | | 0.063 | | | | | |

Only items with Differential Item Functioning are mentioned. All other items had no Differential item Functioning.

negligible. Of the items that were slightly modified between the hospitals (GA2 and GA3), only GA2 was flagged for DIF regarding the sample hospital, but its impact was negligible. A more detailed exploration of DIF within the scales is shown in online Supporting Information S2 Document in S1 File.

**4. Information.** The Test Information Curves and SE(theta) for each scale are shown in Fig 3. The order of the scales regarding difficulty (i.e. the theta's at which the scale is most informative) was from most difficult to least difficult: Regional anaesthesia, Epidural anaesthesia, Spinal anaesthesia, PSA, General anaesthesia I, General anaesthesia II, and Generic items.

## Final questionnaires

The resulting scales from phase I and II each form a separate questionnaire within the RAKQ. S4 Table shows the English translation of these seven final questionnaires on preoperative knowledge on anaesthesia techniques.

## Discussion

This is the first elaborative effort to develop a validated questionnaire that covers the main domains of patient knowledge in anaesthesia in adult care and to evaluate its psychometric properties. The RAKQ is a set of seven questionnaires: one generic questionnaire and six questionnaires covering five different anaesthesia techniques, of which general anaesthesia is divided into two questionnaires. In total the RAKQ contains 40 multiple-choice items. Through a psychometric evaluation, we created unidimensional questionnaires and provided IRT models for each scale.

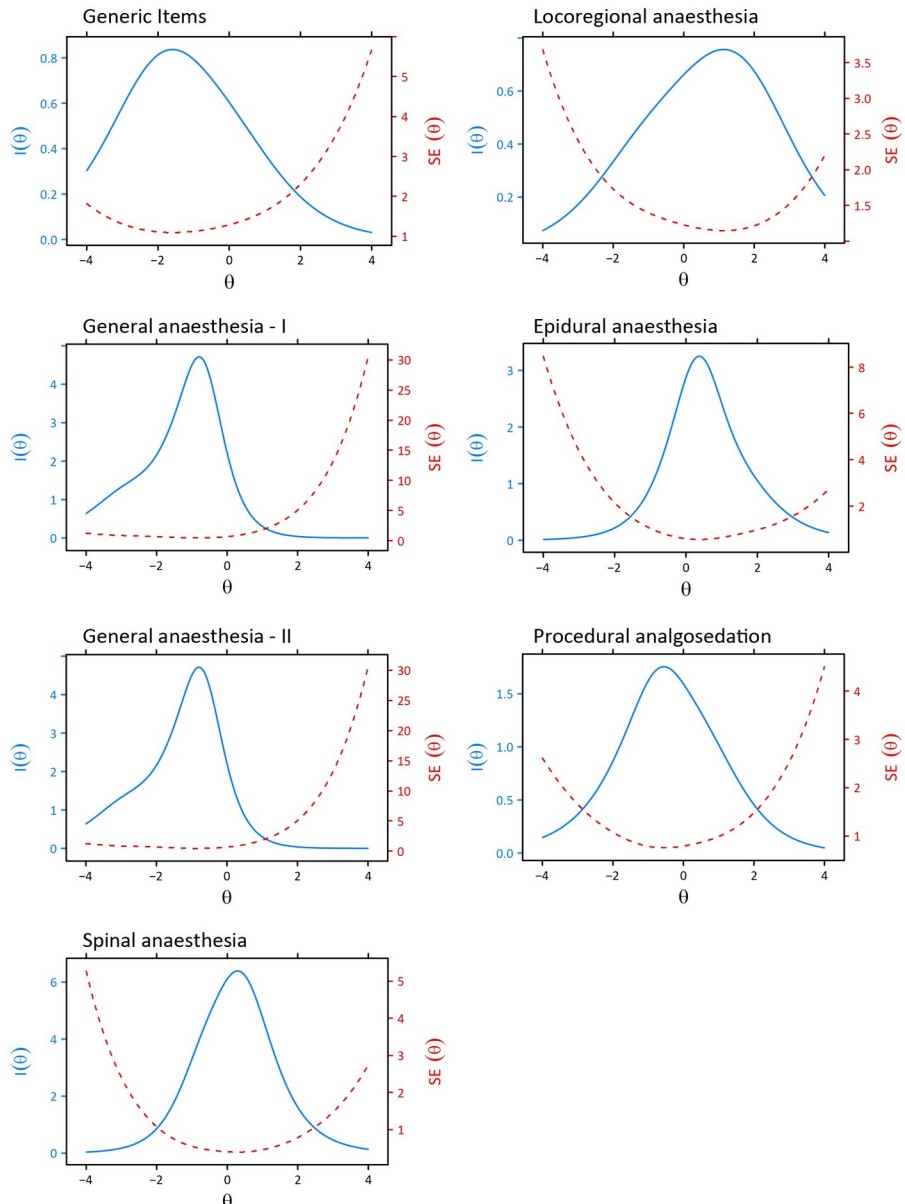

**Fig 3. Test Information (I) and Standard Errors (SE) Curves of the seven scales.**

Several questionnaires have been developed to measure knowledge on anaesthetic topics [2–5] in the past. However, the RAKQ is the first questionnaire to cover all clinically relevant anaesthesia techniques, such as spinal, epidural, and regional anaesthesia, and PSA. Furthermore, the RAKQ has undergone extensive psychometric evaluation, using IRT for the first time in a preoperative knowledge questionnaire. Trustworthy alternatives to in-person patient education on all topics of anaesthesia can only be developed using a validated instrument, such as the RAKQ.

## Methodological considerations

A deductive approach was used to ensure content and face validity in the item development phase (Phase I). This implies that the generation of items was based on predefined domains

formulated by experts. Since this was a first endeavour in constructing a comprehensive knowledge questionnaire on anaesthesia, we conducted exploratory factor analysis to explore underlying latent constructs which could have been less obvious at forehand. During exploratory factor analysis we found some items deemed important by experts had low factor loadings ($<|0.4|$). This could mean that indeed different underlying constructs can be identified, hindering our efforts to create unidimensional scales within predefined knowledge domains. Nevertheless, based on discussions within the team and the fact that the data was collected from a large multicentre sample [11], we proceeded with a lower threshold ($>|0.25|$). In future studies, it could be valuable to also include the patient perspective in developing items, which could enhance unidimensionality of the scales. Subsequently, the resulting lists of items were evaluated using confirmatory factor analysis, which confirmed the unidimensionality of the individual scales. The resulting set of questionnaires can inform the clinician about patients' knowledge on these domains, while also helping to shape and refine the constructs.

Scalability is also important during the development of a valid scale using IRT [7]. We found weak to moderate scalability for all scales, indicating how difficult it is to scale knowledge on relatively broad subjects, such as knowledge on anaesthesia techniques. Generating additional items that align with the same construct could improve scalability, but could also result in a division of the current scales into more scales, each focusing on a narrower knowledge domain, rather than the broader coverage provided by the current questionnaires in the RAKQ. This approach would consequently require a larger number of questionnaires to achieve the same level of coverage as the current set of RAKQ questionnaires which would impair user-friendliness. Still, we believe that improving scalability would be the first step in accurately measuring patients' knowledge, since we aim for tailored consultation and education based on the questionnaire.

Given the exploratory nature of our study, we compared the relative fit of 1-, 2- and 3-PL models to assess which model showed the best fit to the data. The 3-PL models never showed to be an enhancement over the 2-PL models, in line with our approach to reduce the element of guessing by adding an additional response option, "I don't know". It should be noted that when employing the questionnaire, a choice must be made between a 1-PL and 2-PL model. This decision should not only be based on the fit of the model to the data, but also on more fundamental considerations. An argument for using a 1-PL model is the premise that every question is of equal importance in estimating the knowledge level.

Conversely, a 2-PL model provides more opportunities to differentiate between patients with sufficient or insufficient knowledge, with the contribution of the items to the estimation of the knowledge level being different.

## Future directions

The RAKQ can be used to assess the level of knowledge of patients regarding the perioperative process, risks, and preoperative preparations in both clinical practice and research settings. When applied after digital education and before consultation, consulting anaesthetists can pay particular attention to gaps in knowledge. Moreover, patients can automatically be offered additional information even before preoperative consultation. Furthermore, when a fully digital preoperative screening is considered for selected (low-risk) patients, the RAKQ can indicate whether patients have truly understood the information or whether additional education is needed for a well-informed consent. The RAKQ can be used as a research tool to compare new methods of patient education with traditional methods. The practical application of the RAKQ and its acceptance by patients and anaesthetists as a tool to optimise preoperative education should be evaluated in clinical settings.

As a next step, a computerised-adaptive test (CAT) can be developed with the difficulty and discriminatory parameters provided by IRT modelling. With a sufficiently large item pool with items that are sufficiently discriminatory and differ across their difficulty levels, CAT can be used to reliably measure the level of knowledge with a minimal, individualised subset of items to reduce the burden on patients. As can be deduced from the Information Curves, our questionnaires differ in their overall difficulty. The item sets require further diversification regarding the difficulty of the items to facilitate meaningful CAT, such that better discrimination between an acceptable and unacceptable level of knowledge is possible. Furthermore, conducting reliability testing on the validated questionnaires before developing a CAT is warranted to ensure the stability and precision of the IRT parameters across different ability levels. Additionally, targeting populations educated on the specific anaesthesia technique assessed in each questionnaire and providing standardized instructions to the anaesthetist educating the patients, would enhance the development of the questionnaires and, consequently, of the CAT. This approach would results in a more sensitive and reliable assessment of the participants' abilities within the respective domains.

In summary, the set of questionnaires presented in this paper is the first to cover multiple commonly used anaesthesia techniques and the first to be psychometrically validated using IRT. We believe that these questionnaires are a solid foundation on which to further develop knowledge scales and explore computerised-adaptive testing. This can pave the way for trustworthy digital informed consent, which could reduce patient burden and optimise the efficiency of preoperative care.

## Supporting information

**S1 Fig. Scree plots following modified parallel analysis.**
(PDF)

**S1 File. Alterations in the items between the two study sites.**
(DOCX)

**S2 File. Differential item functioning.**
(DOCX)

**S1 Table. Glossary of terms.**
(DOCX)

**S2 Table. Comparison of nested multidimensional Item response theory models for exploratory factor analysis.**
(DOCX)

**S3 Table. Comparison of 1-, 2- and 3-PL item response theory models.**
(DOCX)

**S4 Table. Final questionnaires.**
(DOCX)

**S5 Table.**
(DOCX)

## Acknowledgments

The authors would like to thank Dr. M. Vereen and Dr. E. Galvin for the English translation of the RAKQ.

## Author Contributions

**Conceptualization:** Sander F. van den Heuvel, Sanne E. Hoeks, Sohal Y. Ismail, Jan J. van Busschbach, Robert Jan Stolker, Jan-Wiebe H. Korstanje.

**Data curation:** Sander F. van den Heuvel, Hester van Eeren, Sanne E. Hoeks, Anna Panasewicz, Jan-Wiebe H. Korstanje.

**Formal analysis:** Sander F. van den Heuvel, Hester van Eeren, Sanne E. Hoeks, Sohal Y. Ismail, Jan-Wiebe H. Korstanje.

**Funding acquisition:** Sander F. van den Heuvel, Jan-Wiebe H. Korstanje.

**Investigation:** Sander F. van den Heuvel, Sanne E. Hoeks, Anna Panasewicz, Jan-Wiebe H. Korstanje.

**Methodology:** Sander F. van den Heuvel, Hester van Eeren, Sanne E. Hoeks, Sohal Y. Ismail, Jan J. van Busschbach, Jan-Wiebe H. Korstanje.

**Project administration:** Jan-Wiebe H. Korstanje.

**Resources:** Jan-Wiebe H. Korstanje.

**Software:** Sander F. van den Heuvel.

**Supervision:** Sanne E. Hoeks, Sohal Y. Ismail, Jan J. van Busschbach, Robert Jan Stolker, Jan-Wiebe H. Korstanje.

**Validation:** Hester van Eeren.

**Visualization:** Sander F. van den Heuvel, Hester van Eeren.

**Writing – original draft:** Sander F. van den Heuvel, Hester van Eeren.

**Writing – review & editing:** Sander F. van den Heuvel, Hester van Eeren, Sanne E. Hoeks, Anna Panasewicz, Philip Jonker, Sohal Y. Ismail, Jan J. van Busschbach, Robert Jan Stolker, Jan-Wiebe H. Korstanje.

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
