## [Decision Letter · Decision Letter 0]

2 Nov 2023

PONE-D-23-17620Patient knowledge in anaesthesia: psychometric development of the RAKQ – the Rotterdam Anaesthesia Knowledge QuestionnairePLOS ONE

Dear Dr. van den Heuvel,

Thank you for submitting your manuscript to PLOS ONE. After careful consideration, we feel that it has merit but does not fully meet PLOS ONE’s publication criteria as it currently stands. Therefore, we invite you to submit a revised version of the manuscript that addresses the points raised during the review process.

We look forward to receiving your revised manuscript.

Kind regards,

Stefano Turi

Academic Editor

PLOS ONE

“This study was supported by a grant provided by the Dutch Ministry of Economic Affairs to JK (TKI grant No. EMCLSH200009). The sole responsibility for the content of this publication lies with the authors. The funders had no role in study design, data collection and analysis, decision to publish, or preparation of the manuscript.”

“I have read the journal's policy and the authors of this manuscript have the following competing interests: JK is an unpaid medical adviser for NovaCair B.V., a developer of digital preoperative screening software.”

We note that one or more of the authors are employed by a commercial company: NovaCair

Reviewers' comments:

Reviewer's Responses to Questions

**Comments to the Author**

1. Is the manuscript technically sound, and do the data support the conclusions?

Reviewer #1: Partly

Reviewer #2: Yes

2. Has the statistical analysis been performed appropriately and rigorously? 

Reviewer #1: Yes

Reviewer #2: Yes

3. Have the authors made all data underlying the findings in their manuscript fully available?

Reviewer #1: Yes

Reviewer #2: Yes

4. Is the manuscript presented in an intelligible fashion and written in standard English?

Reviewer #1: Yes

Reviewer #2: Yes

5. Review Comments to the Author

Reviewer #1: Dear authors,

Overall, I find the manuscript may benefit from rearrangement and reanalysis. I included and pointed out specific parts for the comments in the included annotated PDF. In general, please consider the following points:

- The study can be better described by Phase 1 - Scale development and Phase 2 - Scale evaluation/validation. Scale and item development must be developed together, not separately.

- EFA was used in this study to determine suitable latent dimensions from the data. In my opinion, for knowledge scale, this is redundant. If the items were developed while keeping in mind the scale / concepts they are supposed to represent, unidimensionaly will be quite obvious and any items that do not fit in IRT model can be removed from the scale during IRT.

- For the IRT analysis, the sub-questionnaires should be decided based on qualitative and theoretical standpoint, not by EFA findings.

- I find the decision to test out and mix different IRT models difficult to justify in this study. The decision must be based on why you want to use such model (ie to determine

difficulty only assuming same discrimination etc), not because the model fit the data.

- The manuscript will be more representable if it is re-written and reanalyzed. Please consult checklists, for example from Equator network.

- S1 & S4, in my opinion, should be in the main manuscript.

Reviewer #2: The authors discuss the development of a comprehensive questionnaire covering six distinct knowledge domains in anesthesia. The article is well written. The method applied is clear, well defined, and appropriate for creating and validating an instrument of this nature. The amount of effort that was undertaken to develop this instrument is apparent. I would like to thank the authors for their effort.

Having reviewed this manuscript in its entirety, I offer the following comments, questions, and points of clarification:

• Suggest adding references for the following:

o Line 55 – “Miller et al.” [3]

o Line 74 – “Dutch Law on Medical Research” (for readers that are not familiar with Dutch regulatory requirements)

o Line 78 – “This study was conducted in compliance with the principles of the Declaration of Helsinki.”

• The number of items selected based on factor loading have been described as both 50 (line 30) and 51 (line 94). Please clarify.

• Lines 115 to 120 – The authors mention to use 10 participants per item as a rule of thumb when performing an IRT modeling, stating “…which was met by including 577 patients visiting the preoperative outpatient clinic.” While the authors state 510 patients need to be included, it appears from Table 4 that 40 questions were included in the final IRT-model (requiring 400 participants). Additionally, there is some discrepancy in the number participating and those that actually returned survey data. In line 117, the authors mention 577 participants are included. In line 250, the authors indicate that 577 participants completed the questionnaire. In line 251, the response rate is listed at 55% and 28% across the two sites. Line 74 indicates that written informed consent was obtained from all subjects. It is unclear if 577 participants consented to receive the questionnaire, or 577 participants returned the questionnaire, given the response rates listed.

o Please clarify how many participants were approached, how many participants were consented to receive surveys, and how many participants completed surveys at each site.

o If 577 participants received the questionnaire and there were response rates of 55% at Erasmus MC (~ 176 participants [0.55*319]) and 28% at ASZ (~73 participants [0.28*258]), please discuss the limitation of not reaching 400 participants needed for the IRT-model.

o Given the comments above, please adjust the language in lines 250 to 251 to reflect the number of participants that returned surveys; if applicable.

• Lines 122 to 125 – The authors indicate that the questionnaire was adjusted after administration to Erasmus MC participants and prior to administration in ASZ participants. Please provide information on the exact changes that were made (minimally in the supplement). It is unclear to the reader if these changes are significant enough to disrupt the homogeneity of the sample across two groups. Additionally, were any of the changes made presented to the original group (Erasmus MC) in a test-retest method to ensure reliability of the instrument? If not, this should be discussed as a limitation.

• Line 371 – The statement “This questionnaire, the RAKQ, contains a set of seven questionnaires covering five different anesthesia techniques and one generic domain…”, is confusing. Is this one questionnaire with seven components/sections, or seven individual questionnaires? Based on previous language I would assume it is 6 sections in which general anesthesia has been divided into 2 components. Please clarify.

• Lines 397 to 403 – The challenges and importance of scalability are discussed. In line 399, the authors write, “Improvements in scalability could result in more items concerning less broad subjects. This would lead to lengthier questionnaires that are less user-friendly…”. This is implying potentially adding more items in the future to improve scalability. This is actually contradictory to the methods described in this manuscript where the authors removed items to increase scalability (lines 294 to 333).

• Line 418 – The authors describe the potential next steps using computerized-adaptive testing (CAT). While the methods used in this manuscript have been appropriate for developing and validating an instrument, no reliability testing has been performed on the current instrument. Additionally, no description of previous reliability testing on individual items/components that comprise the instrument from the original question sources have been mentioned. Please discuss the need to test reliability prior to developing CAT.

• Lines 128 to 129 – The relationship between the education of anesthetic technique and the administration of the developed instrument lacks clarification:

o Where clinicians instructed to educate participants in any systematic way?

o Were all participants educated on all topics that would be collected, including techniques they would not receive?

o Were participants instructed to answer all questions, or was the questionnaire tailored in which participants were instructed to answer only those sections that applied to their anesthetic technique? If the questions were tailored, was an initial question asking the patient what type of anesthetic technique they would be receiving, asked? Doing so would enhance future CAT development, and should be discussed.

• Please provide information on completeness of returned surveys at each site (ideally by anesthetic technique).

• Careful proof-reading and editing of text is needed (ex: “tot” in abstract line 18; typesetting in table 1 for Factor 3 in Generic items).

6. PLOS authors have the option to publish the peer review history of their article (what does this mean?). If published, this will include your full peer review and any attached files.

Reviewer #1: **Yes: **Wan Nor Arifin

Reviewer #2: No

---

## [Author Response · Author response to Decision Letter 0]

19 Dec 2023

As requested, I have made small corrections to meet the PLOS ONE’s style requirements. Furthermore, the following Funding Statement covers all supporting funding for this study: “This study was supported by a grant provided by the Dutch Ministry of Economic Affairs to JK (TKI grant No. EMCLSH200009). The sole responsibility for the content of this publication lies with the authors. The funders had no role in study design, data collection and analysis, decision to publish, or preparation of the manuscript. There was no additional external funding received for this study.” 

Regarding the Competing Interests statement, it is important to clarify that JK's involvement with Novacair was strictly limited to an unpaid advisory capacity for a brief period. At no time was he employed by the company, and the company did not exert any influence over the study design, data collection and analysis, decision to publish, or preparation of the manuscript. Since JK did not have a commercial affiliation with Novacair, I hope the following update suffices: “JK was an unpaid medical adviser for NovaCair B.V., a developer of digital preoperative screening software. This does not alter our adherence to PLOS ONE policies on sharing data and materials.” 

Reviewer #1:

1. Remark: The study can be better described by Phase 1 - Scale development and Phase 2 - Scale evaluation/validation. Scale and item development must be developed together, not separately.

Answer: We agree with your observation that the development of items and scales must be carried out concurrently, where we may have given the impression that the phases are separate. To accurately convey the message that item and scale development are interconnected, we have modified the headings and Figure 1 as you proposed.

2. Remark: EFA was used in this study to determine suitable latent dimensions from the data. In my opinion, for knowledge scale, this is redundant. If the items were developed while keeping in mind the scale / concepts they are supposed to represent, unidimensionality will be quite obvious and any items that do not fit in IRT model can be removed from the scale during IRT.

For the IRT analysis, the sub-questionnaires should be decided based on qualitative and theoretical standpoint, not by EFA findings.

Answer: We appreciate reviewer's insight regarding the implementation of Exploratory Factor Analysis (EFA) in our study to identify latent dimensions, particularly in relation to the knowledge scales. While we acknowledge the validity of this perspective, and on which we agree in other circumstances, we would like to explain our rationale for employing EFA in the current study. Our questionnaire was developed from the perspective of an anaesthetist, keeping in mind the patient perspective, but the questions where not developed solely from the patient's perspective. Although we had a notion of the various domains within our questionnaire from the anaesthetist standpoint, we were uncertain whether they truly aligned with patients' perceptions. Through our analysis, we discovered that unidimensionality was not completely evident, especially for the overall anaesthesia set, which, when divided into multiple questionnaires, in our opinion, would serve as a more robust foundation for subsequent Item Response Theory (IRT) analysis, in which it is approached to take into account the patient perspective more accurately.

We added an extra sentence to the third paragraph in the discussion:

“This implies that the generation of items was based on predefined domains formulated by experts. Since this was a first endeavour in constructing a comprehensive knowledge questionnaire on anaesthesia, we conducted exploratory factor analysis to explore underlying latent constructs which could have been less obvious at forehand. During exploratory factor analysis we found some items deemed important had low factor loadings (<|0.4|). This could mean that indeed different underlying constructs can be identified, hindering our efforts to create unidimensional scales within predefined knowledge domains. Nevertheless, based on discussions within the team and the fact that the data was collected from a large multicentre sample [11], we proceeded with a lower threshold (>|0.25|). In future studies, it could be valuable to also include the patient perspective in developing items, which could enhance unidimensionality of the scales.” 

(line 422-432)

3. Remark: I find the decision to test out and mix different IRT models difficult to justify in this study. The decision must be based on why you want to use such model (ie to determine

 difficulty only assuming same discrimination etc), not because the model fit the data.

Answer: We recognize that in developing a scale based on supposed concepts, those scales should represent the concepts and unidimensionality should be obvious. Furthermore, we subscribe to the remark that when implementing a questionnaire, e.g. when designing a Computerized Adaptive Test (CAT), a theoretical rationale for the choice of the model should be formulated. The 2-PL model provides more opportunities to differentiate between patients with sufficient or insufficient knowledge, with the contribution of the items to the estimation of the knowledge level being different. An argument for using a 1-PL model is the premise that every question is of equal importance in estimating the knowledge level. Given the exploratory nature of this study, we sought to provide information on the quality of the items and different models. When the development of a CAT is undertaken, a choice should be made based on the characteristics of the data as well as on the rationale behind the choice for a 1- or 2-PL model. To convey this message in the manuscript we added the following passage to the discussion:

“Given the exploratory nature of our study, we compared the relative fit of 1-, 2- and 3-PL models to assess which model showed the best fit to the data. The 3-PL models never showed to be an enhancement over the 2-PL models, in line with our approach to reduce the element of guessing by adding an additional response option, “ I don’t know”. It should be noted that when employing the questionnaire, a choice must be made between a 1-PL and 2-PL model. This decision should not only be based on the fit of the model to the data, but also on more fundamental considerations. A 2-PL model provides more opportunities to differentiate between patients with sufficient or insufficient knowledge, with the contribution of the items to the estimation of the knowledge level being different. Conversely, an argument for using a 1-PL model is the premise that every question is of equal importance in estimating the knowledge level.” (lines 448-455)

Furthermore, we added the IRT parameters of both the 1-PL and 2-PL model for each questionnaire in table 5 and rephrased the results section on IRT model fitting as follows:

“Table 5 shows the item parameters for the 1-PL and 2-PL models fitted to the data. S4 Table shows comparisons between the 1-PL, 2-PL, and 3-PL models for each scale. The 1-PL model was the best fitting model for the Generic items, General anaesthesia – II and Regional anaesthesia scales. The 2-PL model was the best fitting model in the General anaesthesia – I, Spinal anaesthesia, Epidural anaesthesia, and PSA scales. Table 5 also shows the item fit per scale for the best fitting model. No item misfit was detected. Fig 2. shows the item characteristic curves (ICC) per scale for the best fitting model.” (lines 368-374)

4. Remark: The manuscript will be more representable if it is re-written and reanalyzed. Please consult checklists, for example from Equator network.

Answer: We were unable to find a suitable reporting guideline within the Equator network. However, we relied on the papers by Boateng et al and Reeve et al to organize and present our findings. (line 76) We consider these publications to be a solid basis in representing and reporting IRT analyses. As mentioned in our answer on remark #1 we did restructure the paper and incorporated important information that was originally submitted as supporting information into the main manuscript. Furthermore, in line with your remark #7 and remark #28 by Reviewer #2, we made small adjustments to the text to facilitate a better understanding of the progress from the initial item set to the 7 final questionnaires. We believe these revisions, coupled with addressing the remarks of both reviewers makes the manuscript more representable. 

“Each selected factor represented a separate scale, corresponding to a separate questionnaire, and was evaluated in the next phase.” (lines 159-160)

“Initially, six knowledge domains were defined: Generic items, General anaesthesia, Spinal anaesthesia, Epidural anaesthesia, Regional anaesthesia and PSA. Then, 51 items covering these knowledge domains were formulated, which were subsequently supplemented by 9 items following the review of a larger group of experts, in total 60 items. The full list of 60 items …” (lines 260-263)

“The RAKQ is a set of seven questionnaires: one generic questionnaire and six questionnaires covering five different anaesthesia techniques, of which general anaesthesia is divided into two questionnaires. In total the RAKQ contains 40 multiple-choice items.” (lines 406-408)

5. Remark: S1 & S4, in my opinion, should be in the main manuscript.

Answer: Both appendices do provide important information for the reader. We have incorporated them into the main manuscript as Table 1 and Figure 2. Thank you for this suggestion.

In-text remarks:

6. Remark: Abstract must include summarized results from EFA and IRT

Answer: The abstract has been revised to include a summary of the results, as you suggested, with the introduction and conclusion having been shortened to adhere to a 300-word limit. 

7. Remark: Sub-questionnaires should be renamed to sub-domains, facets or sections.

Answer: The term “sub-questionnaires” was intended to refer to the 7 questionnaires of the RAKQ, however, its usage was not consistent. Following your recommendation and a thorough examination of the text, we have revised the language throughout the text to achieve greater uniformity in terminology. We aim to use “domain” when discussing the predefined knowledge domains, “factor” when discussing latent variables explaining patterns of correlation, “construct” when discussing the theoretical concept underlying a scale, “scale” when referring to the instrument that is evaluated in Phase II and “questionnaire” when writing about either the initial set of items presented to the participants, or the final products of the process.

8. Remark: Ethical approval should be last paragraph in this section

Answer: We moved the paragraph to the end of the Methods section. (lines 245-251)

9. Remark: The number of items developed in phase I seem to differ according to the methods and results section.

Answer: In the methods section of our manuscript, we utilized the expertise of two researchers/anaesthetists to develop the initial 51 items. This served as the starting point for the development of the scale. In section B, Content Validity, we expanded the pool of experts consulted, which led to the addition of 9 additional items. We acknowledge that this process is not adequately represented in the manuscript and could benefit from further clarification. 

We changed the text in the Results section in the following way: 

“Initially, six knowledge domains were defined: Generic items, General anaesthesia, Spinal anaesthesia, Epidural anaesthesia, Regional anaesthesia and PSA. Then, 51 items covering these knowledge domains were formulated, which were subsequently supplemented by 9 items following the review of a larger group of experts, in total 60 items. “ (lines 260-263)

10. Remark: what is language level B1

Answer: B1 is a language level as classified by the Council of Europe in the Common European Framework of Reference for Languages (CEFR). We have added a reference to this paragraph. (Line 103)

11. Remark: This should be mentioned clearly as sample size determination for the analysis. In addition, this section is more suitable in Scale Evaluation phase. (Line 114)

Answer: Given the fact we did perform EFA and the rule of thumb is to have about 10 completed questionnaires per item, the calculation of participants needed to fulfil this was incorporated into the Scale Development phase. Concurrently addressing remark #26 by reviewer #2, we changed the text in the results and methods section to clarify the number of participants needed given the number of items developed.

In the methods section:

“As a rule of thumb, the sample size for IRT analysis requires ten participants per item [9]. Therefore, the appropriate number of participants was calculated based on the resulting number of items generated during phase I-1. (Item generation).” (lines 113-115)

In the results section:

“Given the fact 60 items were generated during phase I-1 (Item generation), a sample size of 600 participants would be needed, which was approximated by including 577 patients visiting the preoperative outpatient clinic.“ (lines 270-272)

12. Remark: My personal view on this step, is that, this step was redundant, because the knowledge domains were decided earlier on during the development. In IRT context, each of this domain can be checked for the IRT parameters of interest and checked for unidimensionality. Oblique rotation is also not necessary, because each knowledge domain is not meant to be correlated and each knowledge domain is meant to stand alone from the start. (Line 132)

Answer: We addressed this remark answering remarks #2 and #13.

Answer to remark #2: We appreciate reviewer's insight regarding the implementation of Exploratory Factor Analysis (EFA) in our study to identify latent dimensions, particularly in relation to the knowledge scales. While we acknowledge the validity of this perspective, and on which we agree in other circumstances, we would like to explain our rationale for employing EFA in the current study. Our questionnaire was developed from the perspective of an anaesthetist, keeping in mind the patient perspective, but the questions where not developed solely from the patient's perspective. Although we had a notion of the various domains within our questionnaire from the anaesthetist standpoint, we were uncertain whether they truly aligned with patients' perceptions. Through our analysis, we discovered that unidimensionality was not completely evident, especially for the overall anaesthesia set, which, when divided into multiple questionnaires, in our opinion, would serve as a more robust foundation for subsequent Item Response Theory (IRT) analysis, in which it is approached to take into account the patient perspective more accurately.

We added an extra sentence to the third paragraph in the discussion:

“This implies that the generation of items was based on predefined domains formulated by experts. Since this was a first endeavour in constructing a comprehensive knowledge questionnaire on anaesthesia, we conducted exploratory factor analysis to explore underlying latent constructs which could have been less obvious at forehand. During exploratory factor analysis we found some items deemed important had low factor loadings (<|0.4|). This could mean that indeed different underlying constructs can be identified, hindering our efforts to create unidimensional scales within predefined knowledge domains. Nevertheless, based on discussions within the team and the fact that the data was collected from a large multicentre sample [11], we proceeded with a lower threshold (>|0.25|). In future studies, it could be valuable to also include the patient perspective in developing items, which could enhance unidimensionality of the scales.” (line 422-432)

Answer to remark #13: We also kindly refer to our answer on remark #2 and elaborate in line with this answer, when considering the current remark (#13). In the analysis phase, we found that the items, contrary to the initial assumption of strict separation, demonstrated interconnectedness. Consequently, we opted to employ oblique rotation to allow for items to correlate to one another within the

---

## [Decision Letter · Decision Letter 1]

5 Feb 2024

Patient knowledge in anaesthesia: psychometric development of the RAKQ – the Rotterdam Anaesthesia Knowledge Questionnaire

PONE-D-23-17620R1

Dear Dr. van den Heuvel,

We’re pleased to inform you that your manuscript has been judged scientifically suitable for publication and will be formally accepted for publication once it meets all outstanding technical requirements.

Kind regards,

Stefano Turi

Academic Editor

PLOS ONE

Additional Editor Comments :

Dear Dr van den Heuvel, after careful consideration, I am pleased to endorse the publication of your work. Proposed topic is absolutely interesting and methods and results are extensively explained in the manuscript. I apologize for the delay in my choice, but I was able to obtain a second review only by the second reviewer. In my opinion the authors answered correctly all asked questions (all of them, coming both from reviewer 1 and 2), improving the quality of their paper. For this reason and given elapsed time, I decided to accept this work for publication.

As suggested by reviewer 2, I think that an original dutch version of the RAKQ could be added to supplementary material.

Best Regards,

Stefano Turi 

Reviewers' comments:

Reviewer's Responses to Questions

**Comments to the Author**

1. If the authors have adequately addressed your comments raised in a previous round of review and you feel that this manuscript is now acceptable for publication, you may indicate that here to bypass the “Comments to the Author” section, enter your conflict of interest statement in the “Confidential to Editor” section, and submit your "Accept" recommendation.

Reviewer #2: All comments have been addressed

2. Is the manuscript technically sound, and do the data support the conclusions?

Reviewer #2: Yes

3. Has the statistical analysis been performed appropriately and rigorously? 

Reviewer #2: Yes

4. Have the authors made all data underlying the findings in their manuscript fully available?

Reviewer #2: Yes

5. Is the manuscript presented in an intelligible fashion and written in standard English?

Reviewer #2: Yes

6. Review Comments to the Author

Reviewer #2: I appreciate the authors’ careful and thorough reply and consideration to all comments and recommendations. The revisions have significantly improved the quality of this manuscript. I have no additional recommendations beyond one minor suggestion:

-On line 263-265, the authors write: “The full list of 60 items, as a non-validated English translation of the Dutch original,…”. If allowed by the journal, consider including the final questions in Dutch in addition to the current English version in the supplement for completeness, as the questions presented are reported as a non-validated English translation.

7. PLOS authors have the option to publish the peer review history of their article (what does this mean?). If published, this will include your full peer review and any attached files.

Reviewer #2: No

---

## [Editor Report · Acceptance letter]

3 Jul 2024

PONE-D-23-17620R1 

PLOS ONE

Dear Dr. van den Heuvel, 

I'm pleased to inform you that your manuscript has been deemed suitable for publication in PLOS ONE. Congratulations! Your manuscript is now being handed over to our production team.

Kind regards, 

on behalf of

Dr. Stefano Turi 

Academic Editor

PLOS ONE